

# A study on evaluation of english hybrid teaching courses based on AHP and K-means

Hongchun Jia

College of Foreign Languages, Pingdingshan University, Pingdingshan, Henan, China

## ABSTRACT

In hybrid English teaching, there are many courses and various kinds of assessment, which put higher requirements for teachers' accurate and objective curriculum evaluation. This article adopts the clustering method of unsupervised learning to adapt to more data and give the evaluation method a specific generalization ability. A curriculum evaluation system based on AHP and clustering is proposed. Through hierarchical analysis values of online and offline average grades and final offline assessment scores, multiple hierarchical analysis is carried out, and the K-means method is adopted to refine course evaluation, and non-iterative calculation is carried out for non-deterministic numerical data to complete the final assessment of grades. Based on the sample test of the school's data in recent years, this article finds that the proposed method can distinguish different categories of students well, and the absolute error of K-means classification is less than 0.5. The proposed method can ensure the accurate evaluation of colleges and universities and reduce teachers' burden.

## INTRODUCTION

With the progress of science and technology and the continuous development of computer computing ability, the critical role of mobile devices of science and technology is becoming increasingly apparent. Humanity has entered the information and significant data era, and our lives are also moving toward more convenience and wisdom. Similarly, our daily learning methods have undergone significant changes due to this background. In addition to traditional offline learning, rich online resources provide a favorable support platform for improving ourselves. Therefore, when the COVID-19 pandemic is not over, innovative teaching models and an efficient combination of online and offline teaching can achieve twice the result with half the effort (*Ningning, 2021*; *Haoliang & Xuemei, 2020*). As a subject that is at the forefront of teaching reform and needs close cooperation with teaching practice, English education should be more personalized in ordinary training, improve students' listening, speaking, reading and writing abilities in an all-round way, and actively use online means to improve the disadvantages of traditional offline teaching methods such as less time and single mode (*Huiqiong, 2021*). Presently, most research on hybrid teaching focuses on the online and offline teaching mode setting and self-improvement

Corresponding author
Hongchun Jia, 2216@pdsu.edu.cn

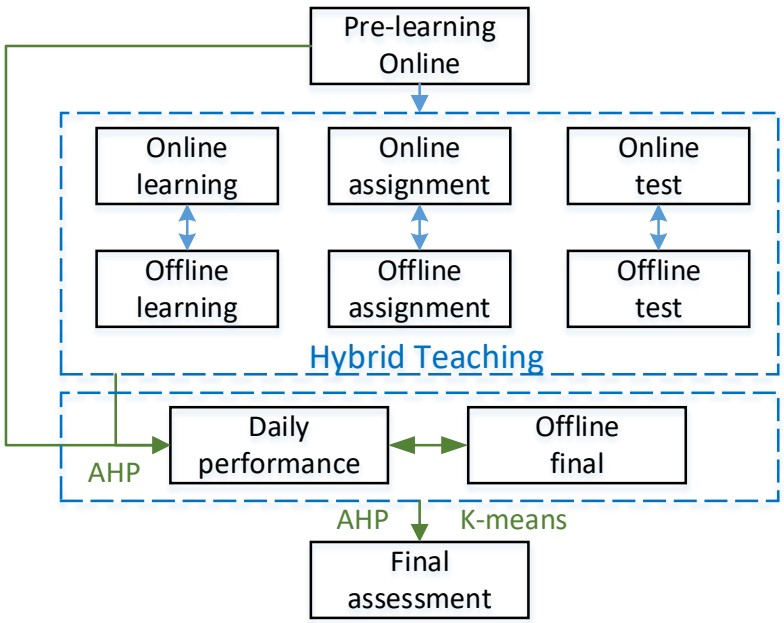

**Figure 1** Evaluation process of hybrid English teaching.

of the course itself (*Xiaocui, 2020*; *Qu, 2020*). The typical education mode of combining online and offline teaching is shown in Fig. 1. Besides that, teachers arrange pre-learning online, and students can do it all by themselves; the rest can be completed through the combination of online and offline teaching, using the hybrid teaching mode. It can be seen that traditional listening, speaking, reading, writing, and other special courses can be taught in this way for English teaching. The hybrid online and offline English teaching mode can fully mobilize students' autonomy in English learning. Online interaction not only improves the efficiency and convenience of teaching but also effectively changes students' psychology of refusing English learning outside the classroom and dramatically increases the interaction between teachers and students, which is very helpful to college students' enthusiasm for English learning (*Xing, 2022*). Although the development of the internet has dramatically improved the efficiency of English teaching, the evaluation process of its teaching effect is very complicated. First, it is difficult to directly reflect online courses' quality and learning effect through the length of students' viewing and online homework. Although various platforms such as MOOCs can supervise through intelligent analysis and task point setting, it is still tricky to intuitively give feedback on the teaching effect. Secondly, due to the curriculum factors, the evaluation needs of different learning stages and other curriculum assignments also increase.

Among various course evaluation systems concerning the online and offline teaching model process described in Fig. 1. *Yan & Tuo (2021)* meticulously divided the process. For example, the evaluation indexes of teaching are divided into modules such as engagement, participation, learning ability and guided assessment and graded with quantifiable indexes such as the number of times students logged in, the length of time they logged in, and

the number of times knowledge points were marked. Finally, the evaluation method, overall grade distribution, and the final grade are given through extensive data analysis methods. Although this method can evaluate the teaching system more comprehensively, it requires a large amount of data and the accuracy of the model will have a significant deviation due to the difference among course settings in different schools. At this stage, the common teaching analysis method is mainly the hierarchical analysis AHP (*Guoyu & Yingchun, 2017*), which decomposes the elements related to decision-making into certain levels, such as goals, criteria, and programs. In teaching, the whole learning process can be explored and objectively evaluated by dividing it into the daily performance, assignment grades and final grades, which correspond to several levels of daily learning, teaching comprehension and integration ability. However, the hierarchical analysis method assumes the independence of each element in the system, which may produce confusion in the real evaluation. To analyze the performance more accurately and objectively, *Yuying (2021)* proposed a network hierarchical analysis model to develop the evaluation indexes of the quality of the hybrid teaching course and completed the evaluation construction of the hybrid course teaching through the improved weight indexes. However, such models require more parameter settings. They are also challenging to apply in practice due to the differences in evaluation systems, curriculum settings, and assessment methods in different schools. *Liu & Li (2020)* proposed an explicit hierarchical clustering of objective functions to improve the solution of objective functions based on global hierarchical clustering. *Hang et al. (2024)* proposed using the analytic hierarchy process (AHP) to evaluate the effective learning of the English teaching language and artificial intelligence and realized that it helps learners choose efficient and effective AI for the English teaching and learning process. Therefore, it is equally essential to ensure the applicability of the proposed method while improving the evaluation efficiency using machine learning and data mining. *Livieris et al. (2019)* proposed a semi-supervised learning algorithm to predict student characteristics to predict student performance on final exams. Starting from the idea of machine learning technology, *Tian (2022)* proposed a teaching evaluation method based on supervised learning to conduct ideological and political evaluations of students. Traditional machine learning methods often form supervised learning models through labeling (*Xun, 2021*), and generalizing supervised models on a large scale is undoubtedly impossible to accomplish for our educational research because the teaching models, such as curriculum and course assessment methods used in each school in English hybrid teaching, are different. There are also considerable differences in the amount of data; therefore, a method with solid generalization ability is needed. Unsupervised learning does not require labeling of training data. The goal of unsupervised learning is to extract useful information and knowledge by exploring the internal structure and patterns of the data. Clustering is a common task in unsupervised learning, which aims to divide data samples into groups or clusters with similar characteristics. The algorithm can help uncover hidden patterns and group structures in the data. A method for evaluating unsupervised teaching models is urgently needed.

This article aims at the common teaching forms of college English teaching and makes the following contributions:

(1) Through hierarchical analysis, the standard scores of the four aspects of English, the regular scores and the final scores of the students were obtained, and multiple layers analyzed the two;

(2) The K-means clustering method is adopted to refine the overall evaluation of students' courses, carry out non-iterative calculation of uncertain values, and complete the objective assessment of teaching results.

## METHODS

### Data of the course AHP analysis

Traditionally, English courses are mainly evaluated in four dimensions: listening, speaking, reading and writing. Most colleges and universities set up their courses in this way, while the final grades are obtained by weighting them through hierarchical analysis. In the original analysis data, firstly, the judgment matrix shown in Eq. (1) is constructed by the weight replication method, where $C_{ij}$ is the scoring system for the importance of elements i and j relative to the upper-level target, and n represents the number of elements, while the matrix is positive definite, diagonal elements are 1 and $C_{ij} = 1/C_{ji}$.

$$C = (C_{ij})_{n \times n}. \tag{1}$$

On this basis, the product $M_i$ of each row of the matrix and its square root $\overline{W_i}$ are calculated as shown in Eq. (2).

$$\overline{W_i} = n\sqrt{M_i}. \tag{2}$$

The vector normalization of Eq. (3) gives the weights of each level.

$$W = \frac{\overline{W_i}}{\sum_{j=1}^{n} \overline{W_j}}. \tag{3}$$

Before calculating the weights, the weights can be set by verifying the maximum eigenvalues of the eigen matrix and then completing the calculation of the consistency indexes to obtain the composite score Y, where $Y_i$ is the index score and n is the number of indexes.

$$Y = \sum_{i=1}^{n} W_i Y_i. \tag{4}$$

Therefore, different schools in the teaching of English for different curriculum settings to complete a more scientific initial evaluation index confirmation of the curriculum; in the author's school, for example, under the influence of the new crown, the school integration of learning pass and its platform including online oral recording homework and usual course practice to form the normal grade, according to the AHP method for the typical grade and the final grade. The primary calculation process of AHP is divided into several processes, including building a ladder hierarchy structure, constructing a judgment matrix, testing the consistency of the judgment matrix, calculating the weight of the algorithm average method, geometric average method and eigenvalue method, filling the matrix, and finally obtaining the result. The steps are as follows: first of all, the four points are included

in the usual grade, then according to the different index weights of the normal and final, to complete a comprehensive assessment.

Given the hybrid teaching mode, the assessment of usual grades may lead to deviations, so the offline final assessment is given higher weight in the actual English course evaluation. Students whose grades are at the edge of passing and the edge of excellence need to be focused on. At the same time, due to the irregularity of the scores, to generalize the evaluation ability, students' grades can be clustered and analyzed to form multiple levels of assessment, such as excellent, medium, and poor, and students at the edge of the clusters are focused on learning re-audit to ensure the objective and accurate evaluation of course learning.

### K-means clustering methods

In English teaching, it is difficult to give a final evaluation directly because of the many different aspects of assessment involved, and it is also difficult to monitor the process by which some students obtain high scores by specific means during online teaching, so bias is inevitable in the evaluation process. Therefore, it is necessary to compare online grades with offline grades and focus on students who have a big difference between them or are at the border of passing or excelling. To adapt to more data and make the evaluation method with a certain generalization ability, this article uses an unsupervised learning clustering method to implement.

Unsupervised learning is a machine learning technique used to find patterns in data using unlabeled data, *i.e.,* only the input variables are given without the corresponding output variables, and the connections that exist within the data are uncovered by the algorithm itself, rather than by learning examples to determine relationships as in supervised learning. Therefore, it is more appropriate to use unsupervised learning to explore the internal linkages in the course evaluation process, where the data are random and difficult to label with large differences from one period to another. In this article, the traditional clustering method in unsupervised learning, K-means, is the most common clustering algorithm based on Euclidean distance, which can be calculated unsupervised and non-iteratively for non-deterministic numerical data. The ultimate goal of the algorithm is to minimize the error square function Eqs. (5) and (6).

$$RSS_k = \sum_{x \in \omega_k} |x - u(\omega_k)|^2 \tag{5}$$

$$RSS = \sum_{k=1}^{K} RSS_k \tag{6}$$

where $\omega_k$ denotes the kth cluster, $u(\omega_k)$ denotes the centroid of the kth cluster, RSSk is the loss function of the kth cluster, and RSS denotes the overall loss function. The optimization goal is to choose the proper attribution scheme to minimize the overall loss function. The specific steps are shown in Algorithm 1.

---

**Algorithm 1**

$\text{K}-MEANS\left(\left\{\vec{x}_1,...,\vec{x}_N\right\},K\right)$

**1** $\left(\vec{s}_1,\vec{s}_2,...,\vec{s}_K\right) \leftarrow \text{SELECTRANDOMSEEDS}\left(\left\{\vec{x}_1,...,\vec{x}_N\right\},K\right)$

**2 for** $k \leftarrow 1$ to $K$
**3 do** $\vec{\mu}_k \leftarrow \vec{s}_k$
**4 while** stopping criterion has not been met
**5 do for** $k \leftarrow 1$ to $K$
**6 do** $\omega_k \leftarrow$
**7 for** $n \leftarrow 1$ to $N$
**8 do** $j \leftarrow \text{argmin}_{j'}\left|\vec{\mu}_{j'} - \vec{x}_n\right|$

**9** $\omega_j \leftarrow \omega_j \cup \left\{\vec{x}_n\right\}$ (reassignment of vectors)

**10 for** $k \leftarrow 1$ to $K$
**11 do** $\vec{\mu}_k \leftarrow \frac{1}{|\omega_k|}\sum \vec{x} \in \omega_k \vec{x}$ (recomputation of centroids)

**12 return** $\left\{\vec{\mu}_1,...,\vec{\mu}_K\right\}$

---

The input of the K-means method is n sample data, $K$-specified clusters, and sometimes a specified number of iterations. Algorithm 1 randomly selects a specified number of K center points and then calculates the distance between each data in the sample and the corresponding center point. In this algorithm, Euclidean distance or cosine distance is generally selected for updating, and the nearest center point is found to realize the return, that is, repeat 5–9 in Algorithm 1. After each class update, when the center of each cluster is determined, repeat 10–11 in Algorithm 1 to make its distribution more uniform. When the termination conditions are met, the cluster analysis is completed. This study selects the iteration times and the change of clustering confidence as the stop conditions. It needs to satisfy formula Eq. (7) and the iterations.

$$\left\|\text{M}(i+1)_{\vec{\mu}_1,...,\vec{\mu}_K} - \text{M}(i)_{\vec{\mu}_1,...,\vec{\mu}_K}\right\|_2 \leq \varepsilon \tag{7}$$

where $\text{M}(i+1)_{\vec{\mu}_1,...,\vec{\mu}_K}$ is the K cluster center position at the i+1 step. The relative error can be obtained by calculating the Euclidean distance between two norms of the results at the ith step, that is, the last time. When the conditions are met, the algorithm stops when the requirements are set and returns to the cluster center position.

## Evaluation system

According to the needs of teaching evaluation under hybrid teaching mode and the characteristics of the methods used, the evaluation process system in this article is shown in Fig. 1.

Given this stage's standard English hybrid teaching mode, this article refines the English teaching evaluation into the details shown in Fig. 1. The blended teaching in Fig. 1 is mainly divided into online pre-learning, blended teaching process, daily performance and final grades, and final assessment results. The specific process is as follows:

1. First, the results are inputted into blended teaching through online pre-learning.
2. Next, English listening and speaking classes can be adjusted according to different teaching modes and times. Online learning can be video, audio recording, and text, and homework can be in various forms, such as submission of audio recordings and manuscripts. Online tests can also be set according to the platform used. Given the differences in the application of different schools, this article only makes a simple introduction.
3. Finally, in the final evaluation stage, two-dimensional K-means cluster analysis is formed according to the currently commonly used combination of peace and final periods, which everyone can intuitively understand. In addition, according to mixed teaching, the score of each part is obtained, and the analytic hierarchy process obtains the usual score. After calculating ordinary grades, the K-means clustering method is used to divide students' grades. Whether there is a big difference between the daily performance and the final score, which affects the grade distribution, we should focus on checking and proofreading.

After completing this step, the analytic hierarchy process is used to weigh ordinary and final offline grades and get the course evaluation. The detailed analysis will be described in the next chapter. In addition to the above roles, we can reduce the burden on teachers by standardizing supervision, inspection, evaluation and assessment matters, standardizing and streamlining report filling, standardizing teacher assessment and training, strengthening organizational leadership, and implementing responsibilities.

## EXPERIMENT RESULT AND ANALYSIS

### Daily performance in hybrid teaching mode

According to the evaluation process shown in Fig. 1, it is necessary to first determine the weight of preview and hybrid teaching through AHP. For the selection of the standard value, 100 is selected considering the future application. In this article, the index weight of each item is calculated according to the situation of the school where the author works, as shown in Table 1:

Considering that there may be some missing parts in the teaching process, it can be adjusted according to practical application needs. For example, online and offline tests or homework can be integrated to form a whole project.

The sample included 119 students from a state higher education school in Peru, randomly selected from different teaching areas (*Guerra Ayala et al., 2023*). To ensure the sample balance and accuracy of data and to test the effectiveness of the designed method based on the extracted data, this study randomly generates a part of the score data through the normal distribution model and finally ensures that the selected data is divided into four groups, and divides them by manual labels, Daily performance is checked by Gaussian distribution to

**Table 1  Evaluation weight of each part under AHP method.**

| Index | Weight | Standard value |
|---|---|---|
| Pre-learning online | 0.10 | 100 |
| Online learning | 0.12 | 100 |
| Online assignment | 0.12 | 100 |
| Online test | 0.12 | 100 |
| Offline learning | 0.18 | 100 |
| Offline assignment | 0.18 | 100 |
| Offline test | 0.18 | 100 |

ensure that they meet $X_{\text{daily performance}} \sim N(60, 20)$, $X_{\text{daily performance}} \sim N(80, 20)$. A normal distribution curve is a probability distribution. A normal distribution is the distribution of a continuous random variable with two parameters $\mu$ and $\sigma^2$, where the first parameter $\mu$ is the mean of the random variable following the normal distribution and the second parameter $\sigma^2$ is the variance of this random variable. The distribution of final grades is similar to the normal distribution of mean and variance. The specific distribution of data is shown in Fig. 2. The reason why we extract the scores of students with such characteristics is that we extract the scores around 60 points in daily performance, 60 and 80 in the final, 80 in daily performance, and 60 and 80 in the final, which ultimately form four groups of students [60,60], [60,80], [80,60] and [80,80]. Their distribution is shown in Fig. 3, and they are classified and marked by corresponding teachers. They are divided into four labels, namely, four clusters, corresponding to Cluster 1: just passed in daily performance and final; Cluster 2: pass in daily performance and excellent in final; Cluster 3: excellent in daily performance and pass in final; Cluster 4: excellent in daily performance and pass in final, a total of four categories. Cluster analysis is selected for such students because it is found in teaching research that such students are on the edge of passing and excellent grades; they are sensitive to grades and need to be carefully reviewed. Therefore, this part of the students was selected as the research focus for the sampling. Before the specific hierarchical cluster analysis, the data needs to be preprocessed. First, the missing value is interpolated, and the unknown value is supplemented with a subjective estimate. Then, the clustering algorithm is used to group the data points. Then, it is determined whether several points do not belong to any cluster to identify the outliers. Finally, some algorithms limit the data from being processed to a specific range.

## Teaching assessment based on the K-means analysis

After the extraction and mixing of the data, the usual and final grades conforming to the distribution law are obtained. Currently, the preliminary analysis of the teaching evaluation grades is completed. Before the final teaching evaluation score is given, a K-means clustering analysis needs to be carried out.

Scatterplot refers to the distribution of data points on the cartesian coordinate plane in regression analysis. The scatterplot represents the approximate tendency of the dependent variable to change with the independent variable, according to which the appropriate logarithm of function data points can be selected for fitting. According to the usual

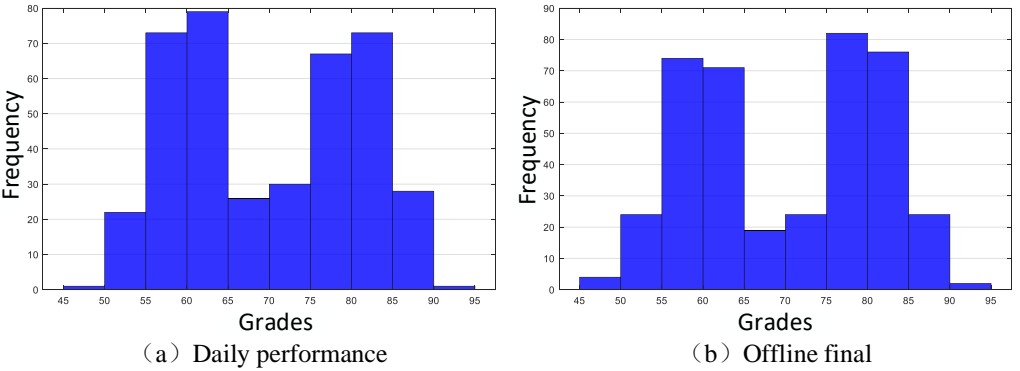

**Figure 2   Distribution histogram of performance evaluation.**

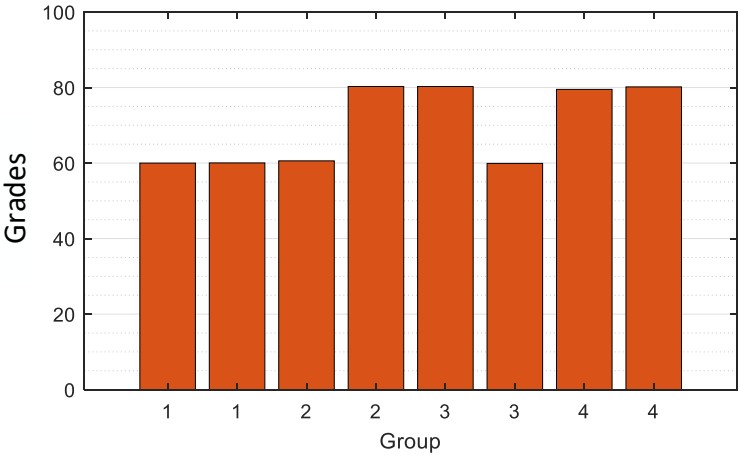

**Figure 3   Distribution of the mean grades in each group.**

scores obtained in the previous section and the scatter plot of the raw data drawn by the corresponding groups, as shown in Fig. 4A, it can be seen that although the teachers who divided the groups were experienced in the description of the data labels, there were still many confusions at the intersection, which also often occurs in the work. Therefore, it is necessary to eliminate such errors through data mining and machine learning, minimize misjudgments, and ensure objectivity and fairness in evaluation.

According to the characteristics of the data used in this article, in addition to setting the number of iterations as the stop condition, this article also uses the threshold method shown in formula Eq. (7) as the stop condition $\varepsilon$, which is set to 0.01. If this is used as the stop condition, the number of iterations will not exceed ten and then stop. To analyze the accuracy of the method, we also conducted experiments on the number of iterations, as shown in Figs. 4B, 4C and 4D. In Fig. 4B, the results of five iterations show that a relatively complete cluster has been formed, but there are some fuzzy classifications at the edges of the two classes. There is almost no difference between the results of multiple iterations of

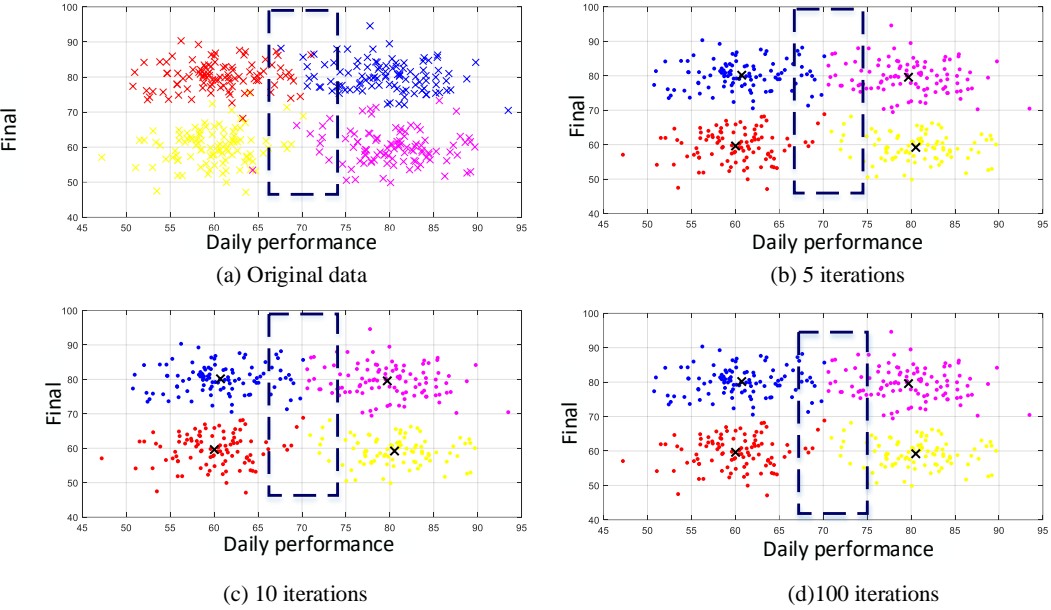

(a) Original data

(b) 5 iterations

(c) 10 iterations

(d)100 iterations

**Figure 4**   **Original data and the clustering in different iterations.**

(c) and (d). When compared with manual labels, the clustering effect is more reasonable. When the number of iterations exceeds 10, K-means can reach a good classification effect. The results in Fig. 4C are the same as those in Fig. 4D, which means that the algorithm converges quickly and accurately distinguishes different categories. It will provide a detailed basis for the final evaluation of teaching results. As seen from the clustering effect diagram shown in Fig. 5, we arranged 400 sample data according to categories and analyzed their automatic clustering results. It can be found from Fig. 4 that more than 15 people in the original sample were wrongly classified, accounting for nearly 4% of the total number, which has a specific deviation for teaching evaluation and needs to be paid attention to in daily teaching and assessment.

In addition, we compared the coordinate points of the cluster center. Due to the fast convergence speed, we found that the coordinate position of the cluster center was fixed within ten times, which is inconvenient. The coordinate of the cluster center is shown in Table 2:

Although there is a specific deviation from the setting average distribution results, the absolute error is less than 0.5, which indicates that the clustering method has strong adaptability to such data and can be competent for evaluating subdivisions.

## DISCUSSION

Online teaching, as a new teaching method, has received widespread attention. Although efficiency and convenience are essential characteristics contributing to its rapid development, through SWOT analysis of online teaching (*Xun, 2021*), it is found that compared with the traditional offline teaching mode, online teaching in the hybrid

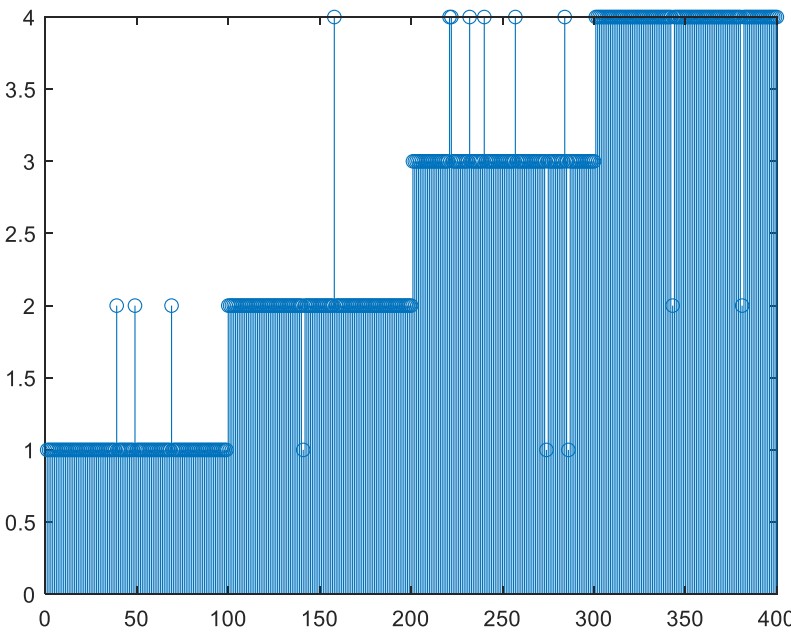

**Figure 5** Comparison between the K-means clustering and original label.

teaching mode requires teachers to design course content, record course videos, and arrange chapter exercises in advance, online tests, related group tasks and other teaching resources record students' learning progress and problems and need to answer questions online. The online and offline hybrid teaching mode requires teachers to spend more time and energy and have a strong sense of responsibility and patience. The improper balance will lead to problems such as excessive workload of teachers. Students' ability to operate and use online platforms is different. Therefore, finding a balance point in the hybrid teaching mode is still necessary. At this stage, due to the impact of the COVID-19 pandemic, online teaching is often carried out passively. Only after the future improvement of curriculum resources and evaluation systems will hybrid teaching improve everyone's enthusiasm for learning and efficiency (*Mengya, 2022*; *Huiyi & Xiangping, 2020*).

The curriculum evaluation system based on hierarchical analysis and clustering method proposed in this article can significantly reduce the pressure on teachers. However, more factors must be referred to for online and offline hybrid courses when giving evaluations, considerably increasing the working time and burden. Fundamentally, hierarchical and cluster analyses ensure accurate and objective assessment. The adopted AHP method is widely used in various evaluations (*Changbo, 2021*; *Guangzhi, 2022*) and performs well in these studies. After clustering analysis, the review can be more objective and accurate. Although the K-means algorithm is sensitive to outliers, its overall variance is slight for students' grades, and the extreme value of the boundary is not essential in the evaluation. At the same time, extreme values can be removed manually first because the assessment focuses on distinguishing students with large deviations in online and offline grades,

| Table 2 Cluster center position. | |
|---|---|
| **Group** | **Center** |
| 1 | [60.02, 59.61] |
| 2 | [60.72, 80.16] |
| 3 | [80.53, 59.20] |
| 4 | [79.70, 79.57] |

not extreme students, so this method has good generalization ability in the curriculum evaluation system.

However, the algorithm's structure shows some limitations, especially in associative cluster analysis. In response to this problem, a future research direction is proposed to improve the effective utilization of the algorithm and its usability in other related fields, which will significantly contribute to improving the quality of the evaluation system and students' learning experience.

## CONCLUSION

Given the complicated evaluation of hybrid online and offline English teaching in universities, this article proposes a curriculum teaching evaluation system based on AHP and the clustering method. The usual grades are calculated using the hierarchical weight of each part of online and offline course teaching, and then the final grades are calculated by AHP again. At the same time, the K-means clustering method is applied to obtain the clustering analysis of student grades. The marginal grades are analyzed and explicitly checked to ensure the course evaluation's objectivity, accuracy, and efficiency. In addition, it can reduce the burden on teachers.

In future studies, this study can refine the analysis of the curriculum system by more detailed segmentation of online and offline teaching results to improve China's mixed teaching mode, stimulate everyone's learning enthusiasm and enhance the evaluation mode.

### Funding
The author received no funding for this work.

### Competing Interests
The author declare that they have no competing interests.

### Author Contributions
- Hongchun Jia conceived and designed the experiments, performed the experiments, analyzed the data, performed the computation work, prepared figures and/or tables, authored or reviewed drafts of the article, and approved the final draft.

## Data Availability

The code is available in the Supplementary Files.

The data is available at Zenodo: Guerra Ayala, M. J., Reynosa Navarro, E., Durand Gómez, E. L., & Apolinar, F. L. (2023). Enjoyment and Development of Oral English Proficiency in Future Teachers [Data set]. Zenodo. https://doi.org/10.5281/zenodo.8401390.

## Supplemental Information

Supplemental information for this article can be found online at http://dx.doi.org/10.7717/peerj-cs.2074#supplemental-information.

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
