# Peer review of "A study on evaluation of english hybrid teaching courses based on AHP and K-means"

_PeerJ Computer Science, doi:10.7717/peerj-cs.2074_

## Round 0.1 · original submission · Major Revisions

Dear Author

Your manuscript has been reviewed by the experts in the field, you will see that they are advising against the publication in its current form and are suggesting a number of improvement comments. Please carefully incorporate these comments along with mine given below.

The paper mentions the higher requirements for accurate and objective curriculum evaluation in hybrid English teaching but does not clearly articulate the specific gap or challenge addressed by the proposed method. Providing a clear statement of the research problem or gap would enhance the paper's clarity and focus

Please define all the acronyms first then use them eg. AHP in abstract and others.

Please Provide a rationale for why unsupervised learning and clustering are suitable for the task at hand, considering factors such as data complexity and scalability, would strengthen the methodological approach.

The paper highlights the success of the proposed method in distinguishing different categories of students and reducing teachers' burden but does not discuss potential limitations or assumptions underlying the findings

Addressing these improvement points would strengthen the paper's contribution to the field of curriculum evaluation in hybrid English teaching and enhance its overall quality and impact. Please provide a detailed rebuttal stating the changes made in light of the comments and their impact in the main updated manuscript file.

Thank you

**Language Note:** PeerJ staff have identified that the English language needs to be improved. When you prepare your next revision, please either (i) have a colleague who is proficient in English and familiar with the subject matter review your manuscript, or (ii) contact a professional editing service to review your manuscript. PeerJ can provide language editing services - you can contact us at [email protected] for pricing (be sure to provide your manuscript number and title). – PeerJ Staff

Reviewer 1 ·

Basic reporting

I have carefully reviewed your manuscript titled “A Study on Evaluation of English Hybrid Teaching Courses Based on AHP and K-means”, and I appreciate the effort put into exploring the assessment of English blended learning courses using intelligent techniques. Below are my comments and suggestions for improving the clarity and robustness of your study:

(1) Moving Figure 1 from the Introduction to the Discussion section would enhance the complexity of the analysis. Figure 1 currently lacks sufficient detail and related descriptions to convey substantial information. Enhancing its complexity and providing additional relevant descriptions would greatly benefit the readers' understanding.

(2) What are the strengths and innovations of this research? Additionally, what potential challenges remain in employing intelligent techniques for evaluating English blended learning courses? The employed technical methods lack novelty, and there is a dearth of extensive literature reviews to demonstrate the feasibility of the proposed approach.

(3) The authors conducted statistical analysis on regular grades and final grades using the AHP method. It is crucial to explicitly define the evaluation criteria used in this analysis, which seems to be lacking emphasis in the manuscript.

Experimental design

(4) Proper data preprocessing, including handling missing values, outlier detection, and data normalization, is indispensable in data mining. The omission of proper data preprocessing might impact the clustering results significantly. It appears that the authors did not adequately address this aspect in their experiments.

(5) As the dimensionality of data increases, the performance of the K-means algorithm may deteriorate due to the ambiguity of distance metrics in high-dimensional spaces. This could result in inaccurate clustering, particularly in course assessments involving numerous features and high-dimensional data. How do the authors plan to mitigate this issue?

(6) I recommend the authors consider integrating feature selection and dimensionality reduction techniques to address the challenges associated with high-dimensional data. Utilizing these techniques can reduce the algorithm's sensitivity to high-dimensional data while improving clustering performance.

Validity of the findings

(7) The manuscript lacks a thorough literature review. A comprehensive literature review should not only list references but also provide in-depth analysis and discussion of viewpoints, methodologies, and conclusions from existing studies. The authors should compare the strengths and weaknesses of different studies, exploring their interrelations and developmental trends.

(8) Given the use of student data for evaluation purposes, addressing ethical considerations, such as data privacy, confidentiality, and consent, would be important for ensuring the ethical integrity of the research and implementation of the proposed system.

(9) Including a section on potential future research directions, such as exploring additional clustering algorithms, integrating qualitative feedback into the evaluation process, or assessing the long-term impact of the proposed system on student learning outcomes, would enrich the discussion and guide future inquiries.

Additional comments

In conclusion, while your manuscript addresses an important topic, it requires significant revisions to strengthen its scientific rigor and contribution to the field. I look forward to reviewing the revised manuscript.

Cite this review as

Reviewer 2 ·

Basic reporting

The introduction adeptly delineates the escalating intricacies inherent in hybrid English teaching and the concomitant evaluation challenges. To further enrich this narrative, consider incorporating specific examples or case studies illustrating these challenges. This augmentation would not only deepen the reader's understanding but also bolster the practical relevance of your study.

While your paper proposes a curriculum evaluation system leveraging AHP and clustering to accommodate the diverse landscape of hybrid English teaching, expounding further on the integration and application of AHP and clustering within the evaluation process would enhance reader comprehension. This transition would seamlessly bridge the rationale behind the chosen methodologies and their operationalization within your study.

Elaborating on the specific evaluation criteria utilized and elucidating the data sources for curriculum evaluation are imperative for gauging the accuracy and reliability of your proposed system. By delving into the rationale behind these criteria and assessing the quality of the data collected, you would fortify the methodological underpinnings of your research.

Experimental design

While the paper mentions conducting sample tests using recent school data to validate the proposed method, amplifying details on the experimental setup—including dataset size, characteristics, and statistical analyses of results—would augment the rigor and credibility of your findings. This transition would seamlessly transition the discussion from methodological description to empirical validation.

While noting the absolute error of K-means classification being less than 0.5, incorporating additional performance metrics such as precision, recall, or F1 score would offer a holistic evaluation of the proposed method's efficacy in student categorization. This transition would facilitate a more nuanced understanding of the method's performance metrics and their implications.
The reduction of teachers' burden, highlighted as a benefit of the proposed method, warrants further elaboration. Expanding on how the system streamlines the evaluation process and alleviates workload pressures for educators would underscore the practical implications and benefits of adopting your curriculum evaluation system. This transition would seamlessly connect the methodological discussion with its practical implications.

Validity of the findings

A discussion on the generalization ability of your proposed method across diverse educational settings and its scalability would provide insights into its broader applicability and potential limitations. This transition would pivot the discourse towards the broader implications of your research beyond the immediate context.

Additional comments

Concluding the manuscript with a succinct summary of key findings, a reaffirmation of contributions, and a reflective commentary on the implications of your proposed curriculum evaluation system for enhancing hybrid English teaching practices would provide a cohesive closure. This transition would seamlessly tie together the various threads of discussion, leaving a lasting impact on the reader.

Cite this review as

---

## Round 0.2 · accepted · Accept

Dear author

Thank you for your submission to our esteemed journal, based on the input from the reviewers on your revised version, I am pleased to accept your article. thank you for your fine contribution.

Reviewer 1 ·

Basic reporting

Paper is in improved shape now

Experimental design

Comments are managed in a effective way

Validity of the findings

After the improvements, paper is in acceptable form

Cite this review as

Reviewer 2 ·

Basic reporting

Responses are satisfactory

Experimental design

ok

Validity of the findings

ok

Additional comments

ok

Cite this review as